# Assessment of Medical Students Burnout during COVID-19 Pandemic

**DOI:** 10.3390/ijerph20043560

**Published:** 2023-02-17

**Authors:** Mohammed A. Muaddi, Maged El-Setouhy, Abdullah A. Alharbi, Anwar M. Makeen, Essa A. Adawi, Gassem Gohal, Ahmad Y. Alqassim

**Affiliations:** 1Department of Family and Community Medicine, Faculty of Medicine, Jazan University, Jazan 45142, Saudi Arabia; 2Department of Community, Environmental and Occupational Medicine, Faculty of Medicine, Ain Shams University, Cairo 11591, Egypt; 3Department of Emergency Medicine, School of Medicine, University of Maryland, Baltimore, MD 21201, USA; 4Department of Surgery, Faculty of Medicine, Jazan University, Jazan 45142, Saudi Arabia; 5Department of Pediatrics, Faculty of Medicine, Jazan University, Jazan 45142, Saudi Arabia

**Keywords:** burnout, medical students, Maslach burnout inventory, COVID-19, environmental health, occupational health

## Abstract

This study estimated the prevalence of burnout and its determinants among medical students at Jazan University during the COVID-19 pandemic. A total of 444 medical students completed an online survey containing the Maslach burnout inventory. The prevalence of burnout was 54.5%. Burnout reached its peak during the fourth year whereas it was the lowest in the internship year. Being a resident in mountain areas, being delayed in college-level, being divorced, and having divorced parents were all associated with an increased risk of burnout. During their time at medical school, students generally showed a trend of consistently high scores in the personal accomplishment subscale, a decreasing trend in the emotional exhaustion subscale, and an increasing trend in the depersonalization subscale. The most important predictive factor was having separated parents. Perceived study satisfaction appeared to be a significant protective factor in a dose–response manner. These findings suggest that burnout among medical students during the COVID-19 pandemic is a concern that should be monitored and prevented.

## 1. Introduction

Burnout syndrome is considered a major public health issue leading to significant physical and emotional impacts [1]. Failure to manage chronic exposure to stress might affect mental well-being and result in burnout syndrome [2,3]. According to Maslach, burnout is characterized by three main components: emotional exhaustion, depersonalization, and reduced personal accomplishment. Emotional exhaustion refers to a feeling of being drained and depleted of energy. Depersonalization is characterized by a negative and detached attitude towards oneself. Reduced personal accomplishment refers to a sense of ineffective and unfulfilling work performance [4]. Burnout can seriously affect a person’s physical, mental, and emotional health and lead to decreased job satisfaction and reduced productivity [5].

Medical students are more vulnerable to developing burnout syndrome than students in other fields due to factors such as high workload, the highly competitive field, a wide range of emotional demands, isolation, and lack of support. [6,7]. Worldwide, they experience substantially higher rates of burnout syndrome [8,9,10]. There are a wide range of reported rates of burnout syndrome in medical students ranging from around 10 to 77% using the Maslach Burnout Inventory Student-Survey (MBI-SS) [11,12,13,14]. It is estimated that around half of medical students might develop burnout at certain points during their time in medical school [15]. The long-lasting effect of future physicians’ burnout is a significant concern that might impact their future careers, leading to an increased rate of medical errors, loss of empathy with patients, poor quality of care, and loss of professionalism [16,17]. It is also a major leading cause of a wide range of physical, mental, emotional, and social consequences impacting the overall quality of life of medical students [16,18]. The COVID-19 pandemic has had a significant impact on the mental and physical well-being of medical students, especially those in clinical years [19]. This is due to several factors, including increased workload, academic stress, future uncertainty, and additional responsibilities and stress brought on by the pandemic [20]. Furthermore, remote learning has made it difficult for medical students to maintain a healthy work–life balance, and the constantly changing situation has heightened stress levels. It is crucial for medical schools and healthcare organizations to prioritize the well-being of medical students and implement strategies to address their risk of burnout [19].

Concerns about these disruptions and adjusting to the new model of teaching brought up by the COVID-19 pandemic might further impact their mental well-being [21]. In this study, we aimed to estimate the burden of burnout and its subscale determinants, and its impacts among medical students at Jazan University in Saudi Arabia. Measuring the magnitude of this issue and understanding its potential consequences and risk factors are crucial for developing effective interventional strategies to mitigate the risk of burnout and its impact on future physicians.

## 2. Materials and Methods

### 2.1. Study Design and Setting

A cross-sectional study was conducted among Jazan University medical students during the 2020–2021 academic year in October–November 2020. The Jazan region lies in the southwestern part of Saudi Arabia and has a population of 1.6 million [22,23]. At the time of data collection, the total number of registered students in the medical school at Jazan University was 941 and the school accepts enrollment of nearly 120 students each year.

### 2.2. Sampling Technique and Study Population

All registered medical students for the 2020–2021 academic year were invited to participate in this study through an online self-administered questionnaire. In addition, the research team conducted several class visits to remind and encourage students to participate in the study. The medical education system in Saudi Arabia follows a six-year traditional curriculum, which consists of a preparatory year of natural sciences and English language courses, two years of basic medical science courses, and three years of clinical training, followed by a one-year internship [24]. We excluded preparatory year students and those who submitted an incomplete MBI-SS section of the questionnaire.

### 2.3. Study Tool

A multi-component online questionnaire was used to collect data for this study. All students received a web link (Google Forms) through WhatsApp. The questionnaire was prepared in Saudi Arabic language and consisted of three sections. The cover page of the questionnaire included the general goals of the study, information about the research team, and the consent form indicating the right of participants not to participate at all or to withdraw at any time without any consequences.

The first section of the questionnaire contained questions on demographic characteristics, including the status of academic progress, average length (in hours) of the school day, and the average duration of studying hours after school.

The second section contained questions on mental health and future career plans: medication history for depression, anxiety, and panic attacks, the academic year of diagnosis (if applicable), interpersonal aggression, perceived study satisfaction, and the intention to leave the school the following year.

The third section of the questionnaire contained the Arabic translated version of the MBI-SS [25]. The MBI-SS is designed to assess 22 items in three subscales of burnout syndrome in students: emotional exhaustion (9 items), depersonalization (5 items), and personal accomplishment (8 items). For both the emotional exhaustion and depersonalization subscale, higher mean scores correspond to a higher degree of experienced burnout. In contrast, lower mean scores on the personal accomplishment subscale correspond to a higher degree of experienced burnout. The items are written in the form of statements about personal feelings or attitudes and are answered in terms of the frequency with which the respondent experiences these feelings on a 7-point scale (ranging from 0, “never”, to 6, “every day”). Each respondent’s test form is scored using a scoring key containing directions for scoring each subscale. The scores for each subscale are considered separately and are not combined into a single total score; thus, three scores are computed for each respondent. High emotional exhaustion was considered at a score of 15 or more, high depersonalization was considered at a score of 7 or more, and low personal accomplishment was considered at a score of 22 or less [26]. Burnout is defined as high scores on emotional exhaustion and depersonalization components and a low score on the personal accomplishment component.

Prior to data collection, the questionnaire was piloted on 20 medical students that were not included in the final statistical analysis, and no modifications were needed to be made. The reliability of the questionnaire was calculated using Cronbach’s alpha.

### 2.4. Statistical Analysis

The collected data were analyzed using Microsoft Office Excel 2007 (v12.0, Redmond, Washington, DC, USA) and Statistical Packages for Social Sciences version 22 (IBM SPSS Corporation, Armonk, New York, NY, USA). Burnout subscales were calculated, and study participants were categorized as having burnout or not according to a cut-off point of 9. Descriptive and inferential statistics were calculated. Chi-square was used to test for significant differences in proportions for categorical variables. Multivariate logistic regression was used to test for factors associated with burnout components.

## 3. Results

Nearly forty seven percent (47.2%) of the registered medical students agreed to participate in this study. The Cronbach’s alpha test for the entire MBI scale demonstrated high reliability with a value of 0.85. Burnout subscales were calculated, and study participants were categorized as having burnout or not (Table 1). Based on this categorization, the overall prevalence of burnout was 54.5%. Males (54.0%) and females (55.0%) had almost similar burnout rates. Regarding the academic year, the burnout level was the highest in the fourth year (67.1%) and the lowest in the internship year (36.4%) (Figure 1). Students from mountain areas were characterized by a high burnout level (65.0%). One-year delayed students had a higher level of burnout (66.2%) compared to those who were regular in their academic plan (52.3%). The marital status of students shows that all divorced students were experiencing burnout. According to the marital status of students’ parents, the highest level of burnout was among students with divorced parents (78.9%) or parents who are not living together (77.1%), and the lowest level of burnout was among students having married parents (51.3%). Parents’ jobs were also an influencing factor in the level of burnout. Students with businessman fathers had the highest level of burnout (63.2%) followed by students with governmental employee fathers (56.1%), and the lowest level of burnout was in students with unemployed fathers (42.9%). On the other hand, burnout was higher in the students with private sector employee mothers (83.3%) than students with housewife mothers (53.0%). According to average daily study hours, burnout was higher in students studying for one hour (60.0%) compared to the rest. Students with psychiatric disorders, such as anxiety, depression, or panic attacks, were more likely to experience burnout (68.4%) than other students (54.6%). Burnout was higher in students who were not satisfied with their studying (64.7%) compared to those who were very satisfied with their studying (40.3%). Students who intended to leave the medical college and move to another college had a higher burnout rate (77.8%) compared to those who did not intend to leave the medical college (52.4%).

Table 2 shows the overall prevalence of the different levels of the three components of burnout among the studied students. These subscales (emotional exhaustion, depersonalization, and personal accomplishment) were classified concerning gender and academic year. The prevalence of emotional exhaustion (68.6% in males and 69.3% in females), depersonalization (74.8% in males and 73% in females), and personal accomplishment (90.7% in males and 91.3% in females) were similar in both genders. The highest emotional exhaustion level was reported among third-year medical students (81.8%), while the lowest level was reported in the seventh-year medical students (i.e., interns). The highest level of depersonalization was among the fourth-year medical students (85.7%), while the lowest level was among the second-year students (65.3%). Finally, the personal accomplishment domain was the highest in the fifth year (95.8%), whereas the lowest level was in the third year (85.9%).

Table 3 shows the correlation between factors affecting burnout and its components. Studying satisfaction was a common significant risk factor with an increasing degree of protection by increasing satisfaction dose-dependently for burnout and all its core components. Apart from personal accomplishment, having separated parents was a common significant risk factor for burnout and all other its core components. In addition to having separated parents, being a resident in the mountain area was a significant risk factor for high emotional exhaustion.

The burnout subscales showed drastically different trajectories over academic years (Figure 2). Personal accomplishment showed a high stationary trendline. Emotional exhaustion showed a decreasing trendline, whereas depersonalization showed an increasing trendline.

## 4. Discussion

We included nearly half of the registered medical students in the Jazan Medical School during the school year 2020–2021. The result of the reliability test of the entire MBI scale was high and comparable to the reliability result of a modified version of the inventory conducted in a Jazan population [27]. We detected alarmingly high burnout rates among medical students (54.5%) [8,28]. The high prevalence of burnout among medical students can be attributed to their exposure to several stressors besides the excessive academic load, such as financial difficulties, peer pressure, being away from home, and concerns about their own health (e.g., nosocomial infections) [29]. The burnout rate in our study is relatively lower than the burnout rate (60.2%) found in a previous study on the same study population during the academic year 2017–2018 using a different scale (i.e., Copenhagen Burnout Inventory) [30]. This could be attributed to the reduction in workload and stress caused by the temporary reduction of clinical rotations by the university [31,32]. More importantly, previous studies show that the Copenhagen Burnout Inventory tends to show relatively higher rates of burnout compared to the MBI [33].

The most important predictive factors of burnout in our study were being a resident in a mountain area and having separated parents. Perceived studying satisfaction was a significant protective factor against burnout in a dose–response manner. Moreover, an inverted-U-shaped relationship was observed between the prevalence of burnout syndrome and the academic year of medical students (Figure 1). As in previous studies, the prevalence of burnout peaks during the transition from pre-clinical to clinical years, probably because of the difficulties in adjusting to the new educational environment as clinical-based teaching with a sudden increase in the academic and clinical requirements [13,34,35,36,37,38]. Other studies have identified the probable sources of the increased psychological impact during the transition period in medical school: uncertainty about their roles, lower than needed supervision, and abrupt increase in workload [39,40,41]. The following decline in the prevalence of burnout during the subsequent years supports this hypothesis.

Medical students must pass their internship year to graduate and fulfill the medical degree requirements. However, during this year, interns get paid a compensation that is about 10 ten times their compensation during the previous years; the year requires no examinations or homework, and they shadow a physician in charge without carrying medicolegal consequences for their clinical decisions regarding patients [42]. All these could explain the marked burnout decline during the internship year. However, burnout decline during internships contradicts the results of another study conducted elsewhere in Saudi Arabia [43]. Our students get their internship training in public MOH hospitals, whereas students in Abdulghani et al. were trained in a university hospital [43]. This discrepancy could be due to the variation in the expected role of medical interns in different training settings. Besides, each school of medicine’s internship unit differs in its proposed services. In the last years, the activation of mentorship and counseling during internships could have lightened the burden of burnout among Jazan medical interns.

In line with other studies, our data suggest that social factors and academic performance play a crucial role in burnout [44]. Being a female, being divorced, having working parents in the private sector, studying for longer hours, and falling behind in their academic plan were found to trigger burnout. In addition, those who are one year delayed in their academic plan showed higher burnout rates than those who were delayed for more than one year. This is probably because of their first trauma experience that seems to recover over time [45]. These findings suggest that it is essential to ensure that medical students get appropriate academic counseling and social support throughout their studies, especially when delayed.

Mountain areas are far (about 100 km) from the college and tend to have poor internet network coverage. Issues with transportation, such as lack of public transportation and not having a personal car, might be reasons for a higher level of burnout among residents of mountain areas. Additionally, as a result of COVID-19, the school was closed, and online classes replaced classroom teaching. Issues with internet connections due to poor network coverage limited the academic performance of those residing in mountain areas and predisposed them to burnout.

Burnout subscales showed different trajectories and trends over the academic years. After passing the first exposure to the medical school’s rich curriculum in the second year, students seem to feel emotional exhaustion and energy depletion. This feeling seems to continue during the transition period (i.e., fourth year) as they are exposed to an even more loaded curriculum and clinical rotations alongside the theoretical part. The expected materials and skills to be mastered in medical school are increasing exponentially [46]. However, during the internship, emotional exhaustion decreases as interns usually relax after finishing most of the requirements and attending the graduation ceremony, and they are well-paid. This subscale mainly mimics the trajectory of overall burnout.

Medical students start idealistic and excited about their acceptance into the medical school, the most competitive school, and their prospective role and social status. However, they tend to get more cynical than other students with time during school studying [47,48]. Although feeling depersonalization is perceived as a negative feeling, acknowledging its existence is a sign of students’ dissatisfaction and indicates that their ideals are still alive [47].

A feeling of reduced personal accomplishment was found in all academic years. This can be attributed to the nature of studying in medical school through high demand and the requirements of persistent studying to comprehend the given materials [49]. Furthermore, although English is the main teaching language, it is the second language for all students. Finally, the need for clinical training and continuing reading and knowledge gained in the clinical years is another factor that may predispose them to always feeling like they are falling behind [46,50].

All these stressors and burnout may impact each other, resulting in mutual negative feedback that might continue through residency and even beyond since a considerable amount of stress will continue to be there [15,51,52]. This vicious circle is highly concerning due to the negative consequences on the mental well-being of future doctors and the quality of healthcare altogether. Therefore, the risk factors of burnout in Jazan university medical students require further systematic investigations. Furthermore, developing screening measures and interventional strategies, especially during the transition from pre-clinical to clinical work, to combat burnout and promote and enhance the ability to cope with stressful lifestyles in our future physicians is a paramount priority to ensure their health and professional well-being. Therefore, we propose integrating these strategies into the medical school curriculum as a component of the medical training based on the weight of evidence about burnout in physicians-in-training.

Burnout was measured with different inventories but MBI is still the widest and most popular scale in use in different languages; however, some publications on the validity of the inventory show controversial results [53,54].

## 5. Limitations

There are several limitations in our study that must be acknowledged. The response rate of interns was very low (2.5%; N = 11) and could not be representative of this group. Additionally, the multifactorial nature of burnout syndrome makes it difficult to track the specific risk factors in our study participants. Finally, despite the widespread use of MBI to measure burnout, further in-depth studies are needed to establish its validity with greater certainty.

## 6. Conclusions

In our study, there was a high prevalence of burnout in Jazan university medical students. Those residing in mountain areas, those who were divorced and children of divorced parents were highly affected by burnout. Study satisfaction is a protective factor against burnout. Our findings can be generalized to all medical students in Saudi Arabia and the Middle East given the similarity in the social and educational circumstances in the region. However, it may not be generalizable to medical students at schools that have had different experiences or during different time frames.

Burnout among medical students is inevitable given the consistent high levels of stress, long hours, and demanding workloads, which can contribute to burnout. However, there are steps that can be taken to mitigate the risk of burnout, such as prioritizing self-care, seeking support from peers and mentors, and engaging in activities that bring contentment. Although burnout is a complex phenomenon and cannot be prevented entirely, it can be mitigated by recognizing and addressing these risk factors and fostering a supportive and healthy culture.

Future studies on burnout syndrome should aim to replicate this study in other medical schools and at different times to determine the generalizability of the findings and to identify factors that may be unique to different populations of medical students. Furthermore, studies can be carried out to explore the effectiveness of various interventions for reducing burnout in medical students and to establish best practices for promoting their well-being.

Overall, the results of this study provide valuable insights into the experiences of medical students and highlight the need for ongoing research and the development of tailored intervention strategies to enhance the well-being of future physicians.

## Figures and Tables

**Figure 1 ijerph-20-03560-f001:**
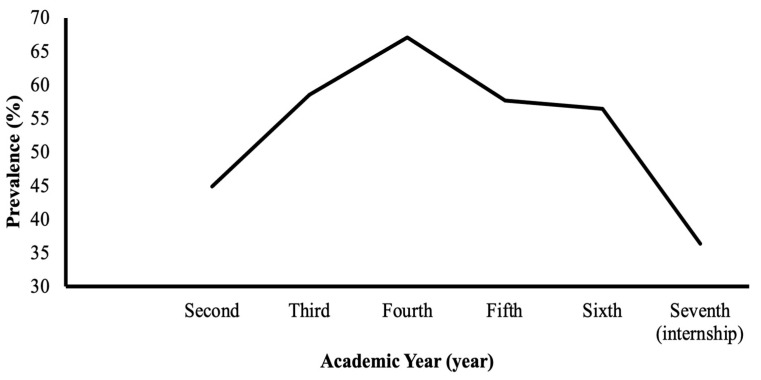
Prevalence of burnout per academic year in medical school among study participants. Participants were classified as burned-out and not burned-out according to a cut-off score of 9.

**Figure 2 ijerph-20-03560-f002:**
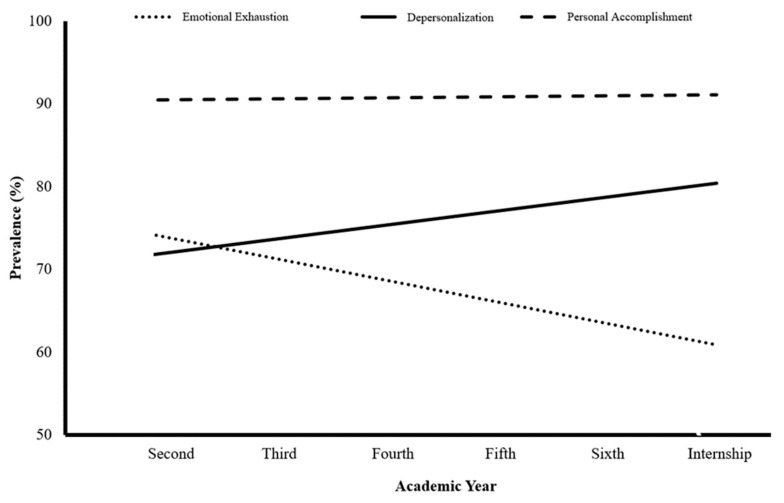
Trendline for the prevalence of high burnout subscale according to the academic year.

**Table 1 ijerph-20-03560-t001:** Categorization of burnout according to selected students’ characteristics.

Factor	Categories	Burnout * N (%)	No Burnout N (%)	Total N
Academic year	Second	66 (44.9)	81 (55.1)	147
	Third	58 (58.6)	41 (41.4)	99
	Fourth	47 (67.1)	23 (32.9)	70
	Fifth	41 (57.7)	30 (42.3)	71
	Sixth	26 (56.5)	20 (43.5)	46
	Seventh (internship)	4 (36.4)	7 (63.6)	11
Progress in the study	Not delayed	161 (52.3)	147 (47.7)	308
	Delayed one year	51 (66.2)	26 (33.8)	77
	Delayed 2 years	14 (45.2)	17 (54.8)	31
	Delayed >2 years	16 (57.1)	12 (42.9)	24
Marital status of the students	Single	224 (53.8)	192 (46.2)	416
	Married	14 (58.3)	10 (41.7)	24
	Divorced	4 (100.0)	0 (0.0)	4
Marital status of the students’ parents	Married and living together	200 (51.3)	190 (48.7)	390
	Married but not living together	27 (77.1)	8 (22.9)	35
	Divorced	15 (78.9)	4 (21.1)	19
Study satisfaction	Very satisfied	50 (40.3)	74 (59.7)	124
	Satisfied	115 (57.2)	86 (42.8)	201
	Not satisfied	77 (64.7)	42 (35.3)	119
Gender	Male	122 (54.0)	104 (46.0)	226
	Female	120 (55.0)	98 (45.0)	218
Residential area	Mountains	13 (65.0)	7 (35.0)	20
	Urban	101 (56.4)	78 (43.6)	179
	Rural	128 (52.2)	117 (47.8)	245
Father’s job	Businessman	12 (63.2)	7 (36.8)	19
	Governmental employee	115 (56.1)	90 (43.9)	205
	Private sector employee	5 (50.0)	5 (50.0)	10
	Unemployed	9 (42.9)	12 (57.1)	21
	Retired	101 (53.4)	88 (46.6)	189
Mother’s job	Businesswoman	2 (66.7)	1 (33.3)	3
	Governmental employee	81 (55.9)	64 (44.1)	145
	Private sector employee	5 (83.3)	1 (16.7)	6
	Housewife	131 (53.0)	116 (47.0)	247
	Retired	23 (53.5)	20 (46.5)	43
Daily hours of study	<1 h	18 (54.5)	15 (45.5)	33
	1 h	21 (60.0)	14 (40.0)	35
	2	42 (46.7)	48 (53.3)	90
	>2 h	161 (56.3)	125 (43.7)	286
Using/used drugs for psychiatric disease	Yes, currently	13 (68.4)	6 (31.6)	19
	Yes, previously	15 (45.5)	18 (45.5)	33
	No	214 (54.6)	178 (45.4)	392
Intention to leave college	Yes	7 (77.8)	2 (22.2)	9
	No	199 (52.4)	181 (47.6)	380
	Maybe	36 (65.5)	19 (34.5)	55
Overall Burnout		242 (54.5)	202 (45.5)	444

* Participants were classified as burned-out and not burned-out according to a cut-off score of 9.

**Table 2 ijerph-20-03560-t002:** The prevalence of burnout syndrome three subscales according to the academic year among medical students (N = 444).

	Emotional Exhaustion			Depersonalization			Personal Accomplishment			Total
Score	(0–9)	(10–14)	(>14)	(0–1)	(2–6)	(>6)	(>27)	(23–27)	(<23)	
	Low N (%)	Mod. N (%)	High N (%)	Low N (%)	Mod. N (%)	High N (%)	Low N (%)	Mod. N (%)	High N (%)	
Academic Year										
Second	37 (25.2)	27 (18.4)	83 (56.5)	5 (3.4)	46 (31.3)	96 (65.3)	1 (0.7)	9 (6.1)	137 (93.2)	147 (100)
Third	10 (10.1)	8 (8.1)	81 (81.8)	4 (4.0)	18 (18.2)	77 (77.8)	10 (10.1)	4 (4.0)	85 (85.9)	99 (100)
Fourth	4 (5.7)	11 (15.7)	55 (78.6)	0 (0.0)	10 (14.3)	60 (85.7)	4 (5.7)	3 (4.3)	63 (90.0)	70 (100)
Fifth	14 (19.7)	9 (12.7)	48 (67.6)	1 (1.4)	18 (25.4)	52 (73.2)	1 (1.4)	2 (2.8)	68 (95.8)	71 (100)
Sixth	6 (13.0)	6 (13.0)	34 (73.9)	0 (0.0)	12 (26.1)	34 (73.9)	3 (6.5)	2 (4.3)	41 (89.1)	46 (100)
Internship	2 (18.2)	4 (36.4)	5 (45.5)	2 18.2)	0 (0.0)	9 (81.8)	0 (0.0)	1 (9.1)	10 (90.9)	11 (100)
Total	73 (16.4)	65 (14.6)	306 (68.9)	12 (2.7)	104 (23.4)	328 (73.9)	19 (4.3)	21 (4.7)	404 (91.0)	444 (100)

Abbreviations: Mod.: Moderate.

**Table 3 ijerph-20-03560-t003:** Logistic regression for factors affecting burnout and its components.

Factor	β	SE	*p*-Value	Odds Ratio (95% CI)
Severe emotional exhaustion (vs. mild/moderate)				
Residence (Mountain)	1.38	0.65	0.035 *	3.96 (1.10–14.21)
^^^ Separated parents	1.97	0.54	<0.001 *	7.18 (2.49–20.70)
^#^ Satisfied	−0.62	0.29	0.033 *	0.54 (0.30–0.95)
^#^ Very satisfied	−1.66	0.31	<0.001 *	0.19 (0.10–0.35)
Constant	1.40	0.24	<0.001 *	
Severe depersonalization (vs. mild/moderate)				
^^^ Separated parents	1.94	0.61	0.001 *	6.95 (2.12–22.82)
^#^ Satisfied	−0.59	0.30	0.048 *	0.56 (0.31–1.00)
^#^ Very satisfied	−1.00	0.31	0.001 *	0.37 (0.20–0.68)
Constant	1.47	0.25	<0.001 *	
Severe personal accomplishment (vs. mild/moderate)				
^#^ Satisfied	0.59	0.36	0.095	1.81 (0.90–3.64)
^#^ Very satisfied	1.68	0.57	0.003 *	5.35 (1.75–16.31)
Constant	1.72	0.26	<0.001 *	
Burnout				
^^^ Separated parents	1.18	0.35	0.001 *	3.26 (1.65–6.43)
^§^ Very satisfied	−0.78	0.22	<0.001 *	0.46 (0.30–0.70)
Constant	0.27	0.12	0.022 *	

β: Regression coefficient. SE: Standard error. CI: Confidence interval. * Significant (*p*-value < 0.050). ^^^ Separated = divorced or married but not living together. ^#^ Compared to non-satisfied. ^§^ Compared to non-satisfied and satisfied.

## Data Availability

Data are available upon request due to ethical restrictions regarding participant privacy. Requests for the data may be sent to the corresponding author.

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
