# Peer review of "Assessment of Medical Students Burnout during COVID-19 Pandemic"

_ijerph, 2023, doi:10.3390/ijerph20043560_

Round 1
Reviewer 1 Report
Thank you for inviting me to review the manuscript. The authors have taken up an interesting and important issue. Noteworthy is an attempt to explore this issue among medical students.
Weaknesses of the manuscript:
- applications not related to the job title and COVID-19, require reformulation
- no psychometric properties of the Maslach Burnout Inventory Student-Survey (MBI-SS) and no psychometric properties during adaptation to the conditions of Saudi Arabia (there is only information about the translation of the MBI-SS into Arabic)
- record of references - standardize the notation: full names or abbreviations of journals
- item 22 raises doubts Maslach, C.; Jackson, S. E.; Leiter, M. P., Maslach burnout inventory: manual. Mind Garden: [Place of publication not identified], 2016.
- Lines 183-185 „The burnout rate in our study is rela- 183 tively lower than burnout rate (60.2%) found in a previous study on the same study pop- 184 ulation using a different scale (i.e., Copenhagen Burnout Inventory)”, when was this study and why, despite COVID-19, in the study using the MBI-SS, the rate of occupational burnout was lower.
I trust that the comments will be used to improve the manuscript, and I congratulate the Authors. I look forward to further research on a large group during the Covid-free period.
Reviewer 2 Report
I suggest that some aspects that were treated superficially in the introduction should be explored in more detail.
Maslach's Burnout model, although interesting and widely recognised, showed several limitations. The authors should at least mention the limitations and advances in the literature. I suggest the following articles:
Schaufeli, W. B., Enzmann, D., & Girault, N. (2017). Measurement of burnout: A review. Professional burnout, 199-215.
Argentero, P., Bonfiglio, N. S., & Pasero, R. (2006). Il burnout negli operatori sanitari volontari. G Ital Med Lav Erg, 28(3), 77-82.
The relationship between covid-19 and mental health is a widely discussed topic and several studies have focused on the effects on students' mental health. The authors should, in my opinion, address this aspect more, analysing its different aspects. I suggest the following articles:
Renati, R., Bonfiglio, N. S., & Rollo, D. (2023). Italian University Students’ Resilience during the COVID-19 Lockdown—A Structural Equation Model about the Relationship between Resilience, Emotion Regulation and Well-Being. European Journal of Investigation in Health, Psychology and Education, 13(2), 259-270.
Panzeri, A., Bertamini, M., Butter, S., Levita, L., Gibson-Miller, J., Vidotto, G., ... & Bennett, K. M. (2021). Factors impacting resilience as a result of exposure to COVID-19: The ecological resilience model. Plos one, 16(8), e0256041.
The authors should also report the months in which the evaluation started and ended, and not only the years
In the description of the Burnout tools, the authors report that they used an Arabic version. They should add more information such as some examples of tem and validity measures such as Cronbach's alpha etc.
It would be better to start the results paragraph with the number 47.2% written in letters
There are, however, some typos errors: "about 10 ten times", "Abdulghani et al. study".
I suggest expanding the conclusions part, writing something more, for instance on the generalisability of some data or bringing in some examples of practical implications of the results. Add a part on the future direction of this research and move the limitations from discussions to conclusions.
